# EMPIR: Ensembles of Mixed Precision Deep Networks for Increased Robustness against Adversarial Attacks

**Sanchari Sen**
Center for Brain-Inspired Computing
School of Electrical and Computer Engineering
Purdue University
West Lafayette, IN, USA
sen9@purdue.edu

**Balaraman Ravindran**
Department of Computer Science and Engineering
Robert Bosch Centre for Data Science and AI
Indian Institute of Technology (IIT) Madras
Chennai, TN, India
ravi@cse.iitm.ac.in

**Anand Raghunathan**
Center for Brain-Inspired Computing
School of Electrical and Computer Engineering
Purdue University
West Lafayette, IN, USA
raghunathan@purdue.edu

## Abstract

Ensuring robustness of Deep Neural Networks (DNNs) is crucial to their adoption in safety-critical applications such as self-driving cars, drones, and healthcare. Notably, DNNs are vulnerable to adversarial attacks in which small input perturbations can produce catastrophic misclassifications. In this work, we propose EMPIR, ensembles of quantized DNN models with different numerical precisions, as a new approach to increase robustness against adversarial attacks. EMPIR is based on the observation that quantized neural networks often demonstrate much higher robustness to adversarial attacks than full precision networks, but at the cost of a substantial loss in accuracy on the original (unperturbed) inputs. EMPIR overcomes this limitation to achieve the "best of both worlds", *i.e.*, the higher unperturbed accuracies of the full precision models combined with the higher robustness of the low precision models, by composing them in an ensemble. Further, as low precision DNN models have significantly lower computational and storage requirements than full precision models, EMPIR models only incur modest compute and memory overheads compared to a single full-precision model ($<25\%$ in our evaluations). We evaluate EMPIR across a suite of DNNs for 3 different image recognition tasks (MNIST, CIFAR-10 and ImageNet) and under 4 different adversarial attacks. Our results indicate that EMPIR boosts the average adversarial accuracies by 42.6%, 15.2% and 10.5% for the DNN models trained on the MNIST, CIFAR-10 and ImageNet datasets respectively, when compared to single full-precision models, without sacrificing accuracy on the unperturbed inputs.

## 1 Introduction

The success of Deep Neural Networks (DNNs) in different machine learning tasks has fueled their use in safety-critical applications like autonomous cars, unmanned aerial vehicles and healthcare, wherein errors (misclassifications) made by DNNs can lead to severe — in the extreme case, fatal — consequences. Therefore, robustness, *i.e.*, the ability to cope with erroneous or malicious inputs fed to an application, is emerging as an important requirement for DNNs.

Several efforts have in fact shown that DNNs behave in unexpected and incorrect ways for small, specifically designed input perturbations (Goodfellow et al. (2014)). An attacker can take advantage of this behavior to intentionally modify the inputs in a manner that forces the DNN model to

mis-classify, and the overall system that uses the DNN to fail. A variety of methods for launching adversarial attacks on DNNs have been proposed over the years. These adversarial attacks systematically modify a given original input to cause a misclassification while keeping the input distortion minimal. A few examples of adversarial attacks that have been successfully applied to various DNN models are the Fast Gradient Sign Method (FGSM) (Goodfellow et al. (2014)), Jacobian-based Saliency Map Attack (JSMA) (Papernot et al. (2015)), Carlini-Wagner (CW) (Carlini & Wagner (2016)) and the Basic Iterative Method (BIM) (Kurakin et al. (2016)).

Prior works have tried to overcome these vulnerabilities by proposing various defense mechanisms against adversarial attacks. Adversarial training (Goodfellow et al. (2014)), defensive distillation (Papernot et al. (2015)) and input gradient regularization (Ross & Doshi-Velez (2017)) are a few representative defense techniques. Each of these approaches, albeit promising, has limitations with respect to the kind of attacks they can defend against, the increase in training complexity, as well as their effect on the model's accuracy on the original unperturbed inputs. To address these shortcomings, we propose EMPIR, an ensemble of mixed precision [1] DNN models, as a new form of defense against adversarial attacks and demonstrate that it can significantly improve the robustness of a variety of DNN models across a wide range of adversarial attacks.

Ensembles have been widely explored as an approach to improve the performance of machine learning models and classifiers (Hansen & Salamon (1990)). Examples of various successful ensembling methods include averaging, bagging (Breiman (1996)), boosting (Dietterich (2000)), *etc*. Recently, it has also been suggested that ensembles may help boost the robustness of DNNs (Strauss et al. (2017); Pang et al. (2019); He et al. (2017); Tramèr et al. (2017)). The individual models in these ensembles are restricted to full precision DNN models, *i.e.*, models utilizing 32 bits of numerical precision to represent different data-structures. Such ensembles are very expensive in terms of the computational and memory overhead (*e.g.*, $10\times$ the baseline for an ensemble with 10 models (Strauss et al. (2017))). In contrast, the use of quantized models in EMPIR, which entail the use of significantly lower number of bits in storage and compute, ensures that the overhead is modest (less than 25% in our evaluations).

Quantized DNNs are characterized by the use of lower numbers of bits to represent DNN data-structures like weights and activations (Venkataramani et al. (2014); Hubara et al. (2017); Zhou et al. (2016); Courbariaux et al. (2015)). They have been widely explored as an approach to reduce the high computational and memory demands of DNNs. Recent studies have also observed that these quantized models demonstrate higher robustness to adversarial attacks (Galloway et al. (2017); Siraj Rakin et al. (2018); Panda et al. (2019)). However, the loss in information associated with the quantization process often makes these quantized models perform significantly worse than their full-precision counterparts while classifying the original unperturbed inputs. This motivates the design of EMPIR, which successfully combines the higher robustness of low-precision models with the higher unperturbed accuracy of the full-precision models. In the general case, EMPIR comprises of $M$ full-precision models and N low-precision models with the final prediction determined by an ensembling technique such as averaging the probabilities or counting the number of predictions for each class. In practice, we find that $M = 1$ and $N = 2$ or 3 provides a significant improvement in adversarial accuracy with small overheads.

In summary, the key contributions of this work are

- We propose the use of ensembles of mixed precision models as a defense against adversarial attacks on DNNs.

- We analyze the effect of ensemble size and ensembling techniques on the overall robustness as well as the computational and storage overheads of the ensemble.

- Across a suite of 3 different DNN models under 4 different adversarial attacks, we demonstrate that EMPIR exhibits significantly higher robustness when compared to individual models as well as ensembles of full-precision models.

---

[1] Here, the term precision refers to the numerical precision of the models, or the number of bits used to represent their weights and activations.

## 2 ADVERSARIAL ATTACKS: BACKGROUND

Adversarial attacks modify inputs in a manner that force a DNN model to misclassify, while ensuring that the input changes are small and imperceptible to human eyes. In the context of DNNs that operate on images, which are the focus of most prior work, various attack methods have been proposed to systematically modify pixel values in the input image so as to result in a mis-classification. A few such methods are described below.

**Fast Gradient Sign Method (FGSM) (Goodfellow et al. (2014)).** FGSM is a single-step attack that operates by calculating the gradient of the loss function with respect to the input pixels ($\nabla_x L(\theta, X, Y)$). Based on the sign of the loss, the input pixels are increased or decreased by a small constant, $\epsilon$, to help move the image towards the direction of increased loss. The adversarial input, $X_{adv}$ can be computed as:

$$X_{adv} = X + \epsilon Sign(\nabla_x L(\theta, X, Y)) \tag{1}$$

Here, $X$ is the original input image associated with an output $Y$ and $\theta$ refers to the weights of the network.

**Basic Iterative Method (BIM) (Kurakin et al. (2016)).** BIM is an iterative version of the FGSM attack which performs a finer optimization by modifying pixels by small values in each iteration. Further, the image generated in each iteration has its pixel values clipped to ensure minimal distortion. Mathematically, this attack can be described as:

$$X_{adv}^0 = X, \quad X_{adv}^{N+1} = Clip_{X,\epsilon}\{X_{adv}^N + \alpha Sign(\nabla_x L(\theta, X_{adv}^N, y))\} \tag{2}$$

Here, the terms $X$, $Y$, $\theta$ and $\epsilon$ have the same meaning as in Equation 1 and $X_{adv}^N$ refers to the adversarial input generated at the $N^{th}$ iteration and $\alpha$ is the step size in each iteration.

**Carlini-Wagner (CW) (Carlini & Wagner (2016)).** CW is another iterative attack that employs optimizers to create strong adversarial inputs by simultaneously minimizing the input distortion and maximizing the misclassification error. It can be described mathematically as:

$$\min_{\delta}\|\delta\|_2^2 + c \cdot f(X + \delta) \text{ such that } (X + \delta) \in [0, 1]^n$$
$$f(X) = \max(\max_{i \neq t}\{Z(X)_i\} - Z(X)_t, 0) \tag{3}$$
$$X_{adv} = X + \delta$$

where $\delta$ is the input distortion, $c$ is the Lagrangian multiplier, $Z(X)$ is the logit output for the input $X$, $t$ is the target class and $f(X)$ is an objective function that satisfies the condition $f(X + \delta) \leq 0$ for all misclassifications.

**Projected Gradient Descent (PGD) (Madry et al. (2017)).** PGD is a third type of iterative attack very similar in nature to the BIM attack. Unlike BIM, which starts with the original image itself, PGD starts with a random perturbation of the original input image. PGD can be described by the following equations:

$$X_{adv}^0 = X + randomUniform(shape(X), \{-\epsilon, \epsilon\})$$
$$X_{adv}^{N+1} = Clip_{X,\epsilon}\{X_{adv}^N + \alpha Sign(\nabla_x L(\theta, X_{adv}^N, y))\} \tag{4}$$

Here, the terms $X$, $Y$, $X_{adv}^N$, $\theta$, $\epsilon$ and $\alpha$ have the same meaning as in Equation 2.

To summarize, different adversarial attacks have been proposed that expose the lack of robustness in current DNN models by constructing adversarial inputs that force a misclassification. Developing defenses to these adversarial attacks is critical to enable the deployment of DNNs in safety-critical systems.

## 3 EMPIR: ENSEMBLES OF MIXED PRECISION DEEP NETWORKS FOR INCREASED ROBUSTNESS AGAINST ADVERSARIAL ATTACKS

To improve the robustness of DNN models, we propose EMPIR, or ensembles of mixed precision models. In this section, we will detail the design of EMPIR models and discuss the overheads associated with them.

### 3.1 Adversarial Robustness of Low-Precision Networks

DNNs have conventionally been designed as full precision models utilizing 32-bit floating point numbers to represent different data-structures like weights, activations and errors. However, the high compute and memory demands of these full-precision models have driven efforts to move towards quantized or low-precision DNNs (Venkataramani et al. (2014); Hubara et al. (2017); Zhou et al. (2016); Courbariaux et al. (2015); Wang et al. (2018)). A multitude of quantization schemes have been proposed to minimize the loss of information associated with the quantization process. While our proposal is agnostic to the quantization method used, for the purpose of demonstration we adopt the quantization scheme proposed in DoReFaNet (Zhou et al. (2016)), which has been shown to produce low-precision models with competitive accuracy values. The quantization scheme can be described by Equation 5.

$$quantize_k(x) = \frac{1}{2^k - 1} round((2^k - 1) \cdot x)$$

$$w_k = 2 \cdot quantize_k(\frac{\tanh(w)}{2 \cdot max(|\tanh(w)|)} + \frac{1}{2}) - 1, \quad a_k = quantize_k(a)$$

(5)

where $k$ refers to the number of quantization bits in the low precision network, $w$ and $w_k$ refer to weight values before and after quantization, and $a$ and $a_k$ refer to activation values before and after quantization.

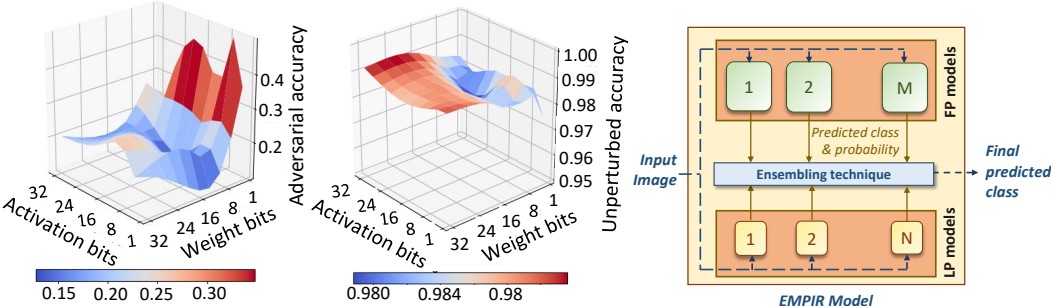

Figure 1: Unperturbed accuracies and adversarial accuracies of low-precision models trained for the MNIST dataset

Figure 2: Overview of EMPIR

In addition to the widely known advantages of reduced model size and reduced complexity of arithmetic computations, recent research efforts have also brought to light another lesser known advantage of low-precision models in the form of increased robustness to adversarial attacks. It has been observed that low-precision models in general exhibit higher values of adversarial accuracy than full-precision models with identical network structures (Galloway et al. (2017); Panda et al. (2019)). One possible explanation for this property is that higher quantization introduces higher amounts of non-linearity, which prevents small changes in the input from drastically altering the output and forcing a misclassification (Galloway et al. (2017)). Figure 1 shows the adversarial accuracies of different low precision models trained on the MNIST dataset under the FGSM attack. Unlike the activations and weights, the gradients utilized in the attack generation process were not quantized, allowing the adversary to launch a stronger attack. From the figure, it is apparent that models with lower numbers of bits used for representing weights and activations exhibit significantly higher levels of adversarial accuracy.

However, increasing the robustness of a system by simply replacing the full-precision model with its low-precision variant can negatively impact its accuracy on the original unperturbed inputs (unperturbed accuracy). In other words, the model may now start to mis-classify inputs that were not adversarially perturbed. Figure 1 also shows the unperturbed accuracies of low-precision models. As expected, models with weights and activations represented using lower numbers of bits exhibit lower unperturbed accuracies.

Based on the above observations, we propose the use of ensembles of mixed-precision models to achieve the best of both worlds, *i.e.*, increase robustness against adversarial attacks without sacrificing the accuracy on unperturbed inputs.

## 3.2 EMPIR: OVERVIEW

Figure 2 presents an overview of EMPIR. In the general case, an EMPIR model comprises of M full-precision (FP) models and N low-precision (LP) models. The full-precision models help in boosting the unperturbed accuracy of the overall model, while the low-precision models contribute towards higher robustness. All the individual models are fed the same input and their predicted classes or probabilities are combined with the help of an ensembling technique at the end to determine the final prediction of the EMPIR model. In practice, we found that a single full-precision model (M=1) and a small number of low-precision models (N=2 or 3) are sufficient to achieve high adversarial accuracies without any noticeable compromises in the unperturbed accuracies.

The ensembling function plays a vital role in the overall performance of the model as it determines the final classification boundary. In this work, we consider two of the most commonly used ensembling functions, namely, averaging and max voting. The averaging function averages the output probabilities of each model and identifies the class with the maximum average probability as the final predicted class. On the other hand, max voting considers the predictions of each model as votes and determines the class with the maximum number of votes to be the final class. In our experiments, we found that averaging achieves better adversarial accuracies on ensembles of size 2 while max voting achieves better adversarial accuracies on ensembles of size greater than 2.

In order to allow an ensemble model to work better than a single model, the individual models should also be designed to be diverse (Hansen & Salamon (1990)). This ensures that the models dont produce similar errors and hence, that the probability of two models misclassifying the same input is lower. We introduce diversity in the individual models of EMPIR by training them with different random initializations of weights.

## 3.3 COMPUTATIONAL AND MEMORY COMPLEXITY OF EMPIR

The ensembling of multiple full-precision and low-precision models in EMPIR increases its computational and storage requirements as these models need to be stored and evaluated. In this work, we keep these memory and computational complexities within reasonable limits by restricting the precision of weights and activations in the low-precision models of EMPIR to a maximum of 4 bits.

The increasing popularity of low-precision DNN models has prompted recent hardware platforms including GPUs and neural network accelerators to add native hardware support for operating on low precision data (Fleischer et al. (2018); Kilgariff et al. (2019)). These hardware platforms reconfigure a common datapath to perform computations on full-precision data (32 or 64 bits) as well as low-precision data (4, 8 or 16 bits). Low-precision operations can achieve higher throughputs than full-precision operations on these platforms as the same number of compute elements and the same amount of memory bandwidth can support a larger number of concurrent operations. Consequently, the additional execution time required to evaluate the low-precision models in EMPIR is much less than that of a full-precision model. Overall, we quantify the execution time and storage overhead of an EMPIR model using the formula described by Equation 6.

$$TimeOverhead\{EMPIR(M, N)\} = M + \sum_{i=1}^{N} \frac{Ops\_per\_sec(FP)}{Ops\_per\_sec(k_i)}$$

$$StorageOverhead\{EMPIR(M, N)\} = M + \sum_{i=1}^{N} \frac{k_i}{FP}$$

(6)

where $k_i$ is the precision of the $i^{th}$ low-precision model, $FP$ is the precision of the full-precision models, and $Ops\_per\_sec(b)$ is the throughput of $b$ bit operations on the underlying hardware platform.

## 4 EXPERIMENTS

In this section, we describe the experiments performed to evaluate the advantages of EMPIR models over baseline full-precision models.

### 4.1 BENCHMARKS

We studied the robustness of EMPIR models across three different image recognition DNNs, namely, MNISTconv, CIFARconv and AlexNet. The individual full-precision and-low precision networks within the EMPIR models were designed to have identical network topologies. The details of the individual networks in these benchmarks are listed in Table 3 within Appendix A. The benchmarks differ in the number of convolutional layers, fully connected layers as well as the datasets. We consider three different datasets, namely, MNIST (Lecun et al. (1998)), CIFAR-10 (Krizhevsky (2009)) and ImageNet (Deng et al. (2009)) which vary significantly in their complexity. The low precision networks were obtained using the quantization scheme proposed in DoReFa-Net (Zhou et al. (2016)). The full precision models were trained using 32 bit floating point representations for all data-structures.

### 4.2 EVALUATION OF ROBUSTNESS

We implemented EMPIR within TensorFlow (Abadi et al. (2015)) and have released the source code for our implementation [2]. The robustness of the EMPIR models was measured in terms of their adversarial accuracies under a variety of white-box attacks within the Cleverhans library (Papernot et al. (2018)). We specifically consider the four adversarial attacks described in Section 2. The adversarial parameters for the attacks on the different benchmarks are presented in Table 1. The attacks were generated on the entire test dataset for each of the benchmarks. Generating these white-box attacks involves computation of the gradient $\nabla_x L(\theta, X, Y)$ (Section 2), which is not directly defined for ensembles. For the EMPIR models, we compute this gradient as an average over all individual models for an averaging ensemble and as an average over the individual models that voted for the final identified class for a max-voting ensemble.

| Network | CW | FGSM | BIM | PGD |
|---------|-----|------|-----|-----|
| **MNISTconv** | Attack iterations = 50 | $\epsilon = 0.3$ | $\epsilon = 0.3$, $\alpha = 0.01$ No. of iterations = 40 | $\epsilon = 0.3$, $\alpha = 0.01$ No. of iterations = 40 |
| **CIFARconv** | Attack iterations = 50 | $\epsilon = 0.1$ | $\epsilon = 0.1$, $\alpha = 0.01$ No. of iterations = 40 | $\epsilon = 0.1$, $\alpha = 0.01$ No. of iterations = 40 |
| **AlexNet** | Attack iterations = 50 | $\epsilon = 0.1$ | $\epsilon = 0.1$, $\alpha = 0.01$ No. of iterations = 40 | $\epsilon = 0.1$, $\alpha = 0.01$ No. of iterations = 5 |

Table 1: Attack parameters

## 5 RESULTS

In this section, we present the results of our experiments highlighting the advantages of EMPIR models.

### 5.1 ROBUSTNESS OF EMPIR MODELS ACROSS ALL ATTACKS

Table 2 presents the results of our experiments across different benchmarks. The EMPIR models presented in the table are the ones exhibiting highest average adversarial accuracies under the constraints of <25% compute and memory overhead and <2% loss in unperturbed accuracy. We observed that across all the benchmarks, ensembles comprised of two low-precision and one full-precision model combined with the max-voting ensembling technique satisfy these constraints. However, the individual configurations of the low-precision models, *i.e.*, the precisions of weights and activations in the ensembles, differ across the benchmarks. For example, both low-precision models in the EMPIR model for MNISTconv have weight precisions of 2 bits and activation precisions of 4 bits. On the other hand, the two low-precision models in the AlexNet EMPIR model have {weight,activation} bit-precisions of {2,2} and {4,4}, respectively. In general, we observe that

---

[2] https://github.com/sancharisen/EMPIR

| Network | Approach | Unperturbed Accuracy (%) | Adversarial Accuracy (%) | | | | |
|---|---|---|---|---|---|---|---|
| | | | CW | FGSM | BIM | PGD | Average |
| MNISTconv | Baseline FP | 98.87 | 3.69 | 14.32 | 0.9 | 0.77 | 4.92 |
| | **EMPIR** | **98.89** | **86.73** | **67.06** | **18.61** | **17.51** | **47.48** |
| | Defensive Distill. | 98.12 | 2.34 | 40.22 | 7.61 | 3.28 | 13.36 |
| | Inp. Grad. Reg. | 99.01 | 6.83 | 30.15 | 1.14 | 1.20 | 9.83 |
| | FGSM Adv. Train | 99.06 | 3.09 | 76.56 | 0.87 | 0.39 | 20.23 |
| | **EMPIR (FGSM Adv. Train)** | **99.09** | **90.54** | **75.98** | **33.16** | **5.17** | **51.21** |
| CIFARconv | Baseline FP | 74.54 | 13.38 | 10.28 | 11.97 | 10.69 | 11.58 |
| | **EMPIR** | **72.56** | **48.51** | **20.45** | **24.59** | **13.55** | **26.78** |
| | FGSM Adv. Train | 72.36 | 14.36 | 41.58 | 12.92 | 11.24 | 20.03 |
| | **EMPIR (FGSM Adv. Train)** | **73.62** | **45.73** | **31.67** | **29.55** | **14.74** | **30.42** |
| | PGD Adv. Train | 73.55 | 12.62 | 12.45 | 10.97 | 8.52 | 11.14 |
| AlexNet | Baseline FP | 53.23 | 9.94 | 10.29 | 10.81 | 10.30 | 10.34 |
| | **EMPIR** | **55.09** | **29.36** | **21.65** | **20.67** | **11.76** | **20.86** |

Table 2: Unperturbed and adversarial accuracies of the baseline and EMPIR models across different attacks

the EMPIR models exhibit substantially higher adversarial accuracies across all attacks for the three benchmarks.

We also compare the benefits of EMPIR with four other popular approaches for increasing robustness, namely, defensive distillation (Papernot et al. (2015)), input gradient regularization (Ross & Doshi-Velez (2017)), FGSM based adversarial training (Goodfellow et al. (2014)) and PGD based adversarial training (Madry et al. (2017)). The distillation process was implemented with a softmax temperature of $T = 100$, the gradient regularization was realized with a regularization penalty of $\lambda = 100$, while the adversarial training mechanisms utilized adversarial examples generated with a maximum possible perturbation of $\epsilon = 0.3$. Table 2 presents the results for the approaches that were able to achieve $<5\%$ loss in unperturbed accuracy for a particular benchmark. We observe that FGSM based adversarial training significantly boosts the adversarial accuracies of the MNIST-conv and CIFARconv models under the FGSM attack but is unable to increase the accuracies under the other three attacks, often hurting them in the process. A similar result is observed for the MNISTconv model trained with defensive distillation and gradient regularization. In contrast, EMPIR successfully increases the robustness of the models under all four attacks. In fact, it can even be combined with the other approaches to further boost the robustness, as evident from the adversarial accuracies of an EMPIR model comprising of adversarially trained models for the MNISTconv and CIFARconv benchmarks. EMPIR also achieves a higher adversarial accuracy than PGD based adversarial training for the CIFARconv benchmark. Overall, EMPIR increases robustness with zero training overhead, as opposed to considerable training overheads associated with the other defense strategies like adversarial training, defensive distillation and input gradient regularization.

## 5.2 COMPARISON WITH INDIVIDUAL MODELS

Figure 3(a) illustrates the tradeoff between the adversarial and unperturbed accuracies of the individual DNN models and EMPIR models for two of the benchmarks under the CW attack. The circular blue points correspond to individual models with varying weight and activation precisions while the red diamond points correspond to the EMPIR models presented in Section 5.1. The figure clearly indicates that the EMPIR models in both the benchmarks are notably closer to the desirable top right corner with high unperturbed as well as high adversarial accuracies. Among the individual models, the ones demonstrating higher adversarial accuracies but lower unperturbed accuracies (towards the top left corner) correspond to lower activation and weight precisions while those demonstrating

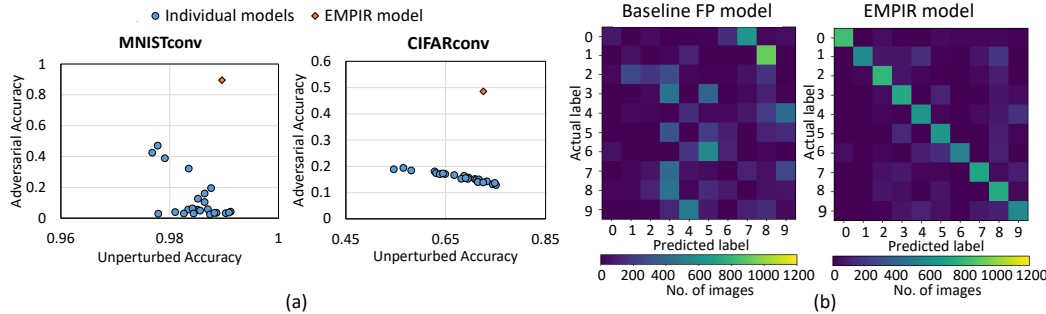

Figure 3: (a) Tradeoff between unperturbed and adversarial accuracies of the individual and EM-PIR models across 2 benchmarks. (b) Confusion matrices of the baseline FP and EMPIR model for the MNISTconv benchmark.

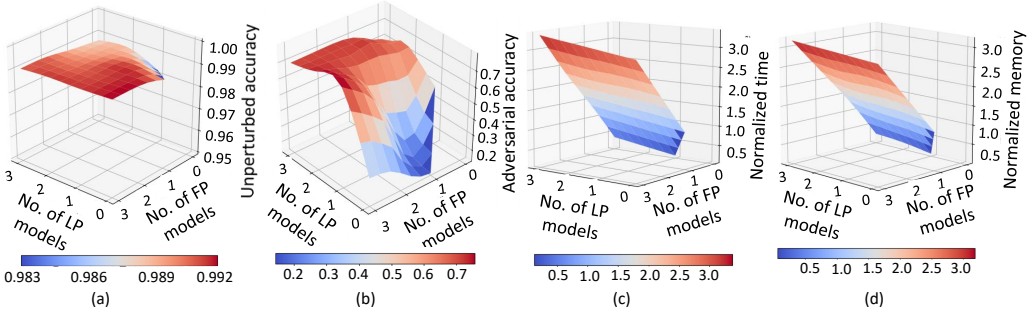

Figure 4: Effects of varying the number of LP and FP models in EMPIR (a) Unperturbed accuracies, (b) Adversarial accuracies, (c) Execution time overheads and (d) Storage overheads

lower adversarial accuracies and higher unperturbed accuracies (towards the bottom right corner) correspond to higher activation and weight precisions.

## 5.3 ANALYSIS OF CONFUSION MATRICES

Figure 3(b) presents the confusion matrices of the baseline FP model and the EMPIR model for the MNISTconv benchmark under the FGSM attack. The actual ground truth class labels are listed vertically while the predicted labels are listed horizontally. The colors represent the number of images in the test dataset that correspond to the combination of actual and predicted class labels. The diagonal nature of EMPIR's confusion matrix clearly illustrates its superiority over the FP model, which frequently misclassifies the generated adversarial images.

## 5.4 IMPACT OF VARYING THE NUMBER OF LOW-PRECISION AND FULL-PRECISION MODELS

In this subsection, we vary the number of low-precision and full-precision models in EMPIR between 0 and 3 to observe its effect on the unperturbed and adversarial accuracies of the MNISTconv benchmark under the FGSM attack. We also measure the execution time and memory footprint of the EMPIR models to quantify their overheads with respect to a baseline single full-precision model. We restrict the low-precision models to have weight and activation precisions between 2 and 4 bits and choose the configurations that maximize the adversarial accuracies of the EMPIR models while introducing <1% drop in unperturbed accuracies.

Figure 4 presents the results of this experiment. Figure 4(a) and (b) clearly indicates that a higher number of low-precision models in EMPIR helps in boosting the adversarial accuracies while a higher number of full-precision models help in boosting the unperturbed accuracies. For instance, an EMPIR model comprising of only three low-precision models demonstrates unperturbed and adversarial accuracies of 98.8% and 56.9% respectively while an EMPIR model comprising of only three full-precision models demonstrates unperturbed and adversarial accuracies of 99.2% and 31%,

respectively. The execution time and memory footprint associated with the former are only $0.38\times$ and $0.25\times$ over the baseline, as opposed to $3\times$ in case of the latter. Overall, we observe that an EMPIR model comprising of a single full-precision model and two low-precision models (configuration presented in Table 2) achieves a good balance between adversarial and unperturbed accuracies with modest execution time and storage overheads.

## 6 RELATED WORK

Popular defense strategies against adversarial attacks include adversarial training, defensive distillation and input gradient regularization. Adversarial training (Goodfellow et al. (2014); Madry et al. (2017)) involves modifying the loss function to include the adversarial loss term, which tries to reduce the effect of input perturbations. Defensive distillation (Papernot et al. (2015)), on the other hand, is based on the technique of distillation that was originally proposed to efficiently transfer knowledge across different DNN models. It involves training networks on the output probabilities of classes instead of the conventional approach of training on hard output class labels. As shown in Table 2, the benefits of these techniques are limited to only one or a couple of adversarial attacks. In contrast, EMPIR is able to boost the adversarial accuracies of DNNs across all four white-box attacks considered here.

Recent efforts have also proposed the use of ensembles of full precision models for defending DNNs against adversarial attacks (Strauss et al. (2017); Pang et al. (2019); He et al. (2017); Tramèr et al. (2017)) However, the presence of multiple full-precision models in these ensembles increases the compute and memory requirements significantly ($10\times$ for an ensemble with 10 models in Strauss et al. (2017)), which prevents the application of this approach to larger state-of-the art models. In contrast, with the use of low precision models in the ensemble, we are able to reduce the overhead significantly and restrict it to $<25\%$. Also, as shown in Figure 4, mixed-precision ensembles demonstrate higher adversarial accuracies than the full-precision ensembles for identical number of models in the ensemble.

In addition to the above efforts, there have been a parallel set of efforts studying the robustness of low-precision or quantized DNNs. For example, binary neural networks with single bit precisions for weights and activations have been shown to exhibit higher adversarial robustness than their full-precision counterparts on different white-box attacks (Galloway et al. (2017); Panda et al. (2019)). Stochastic quantization of activations has also been proposed as an approach to make DNNs more robust (Siraj Rakin et al. (2018)). However, as shown in Figure 1, the individual quantized models in these efforts often demonstrate lower accuracies on unperturbed or clean examples due to the loss in information associated with the quantization process. On the other hand, the combination of full-precision models along with low-precision models in EMPIR helps to overcome this limitation and achieve the best of both worlds — higher robustness combined with high unperturbed accuracy.

## 7 CONCLUSION

As deep neural networks get deployed in applications with stricter safety requirements, there is a dire need to identify new approaches that make them more robust to adversarial attacks. In this work, we boost the robustness of DNNs by designing ensembles of mixed-precision DNNs. In its most generic form, EMPIR comprises of M full-precision DNNs and N low-precision DNNs combined through ensembling techniques like max voting or averaging. EMPIR combines the higher robustness of low-precision DNNs with the higher unperturbed accuracies of the full-precision models. Our experiments on 3 different image recognition benchmarks under 4 different adversarial attacks reveal that EMPIR is able to significantly increase the robustness of DNNs without sacrificing the accuracies of the models on unperturbed inputs.

### ACKNOWLEDGMENTS

This work was supported by C-BRIC, one of six centers in JUMP, a Semiconductor Research Corporation (SRC) program, sponsored by DARPA.

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

# A  BENCHMARK DETAILS

| Network | Dataset | Configuration |
|---|---|---|
| **MNISTconv** | MNIST | Conv(8×8×64), ReLU, Conv(6×6×128), ReLU, Conv(5×5×128), ReLU, Fully_Connected(10), SoftMax |
| **CIFARconv** | CIFAR-10 | Conv(5×5×32), ReLU, MaxPool(3×3), Conv(8×8×64), ReLU AvgPool(3×3), Conv(8×8×64), ReLU, AvgPool(3×3), Fully_Connected(64), Fully_Connected(10), SoftMax |
| **AlexNet** | ImageNet | Conv(12×12×96), ReLU, Conv(5×5×256), BatchNorm, ReLU, MaxPool(3×3), Conv(3×3×384), BatchNorm, ReLU, MaxPool(3×3), Conv(3×3×384), BatchNorm, ReLU, Conv(3×3×256), BatchNorm, ReLU, MaxPool(3×3), Fully_Conn(4096), BatchNorm, ReLU, Fully_Conn(4096), BatchNorm, ReLU, Fully_Conn(1000), SoftMax |

Table 3: Benchmarks

# B  ROBUSTNESS TO ATTACKS OF VARYING STRENGTHS

To further illustrate the benefits of EMPIR, we observed its robustness under attacks of varying strength. We specifically varied the $\epsilon$ value in the FGSM attack between 0.1 and 0.8 and the number of attack iterations in the CW attack between 10 and 90, and measured the adversarial accuracies of the EMPIR model as well as the baseline FP model for the MNISTconv benchmark. Figure 5 clearly illustrates that EMPIR exhibits higher adversarial accuracies across attacks of different strengths for both FGSM and CW.

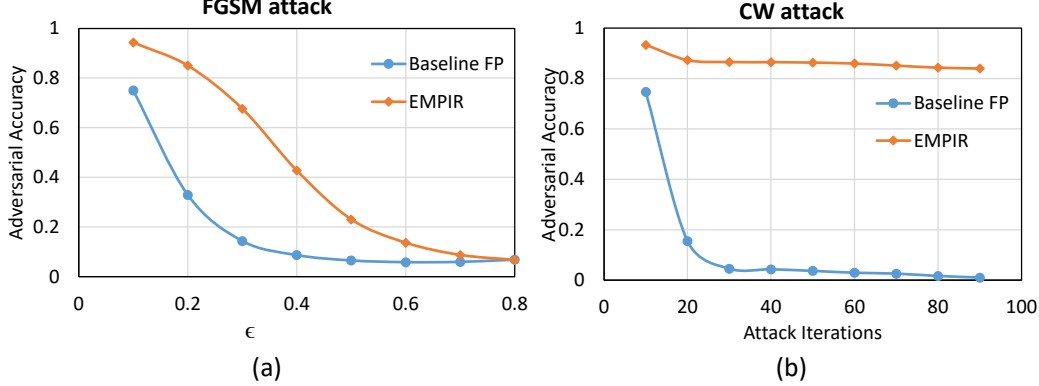

Figure 5: (a) Adversarial accuracies under FGSM attack of varying strength (b) Adversarial accuracies under CW attack of varying strength

