# OpenReview forum: "EMPIR: Ensembles of Mixed Precision Deep Networks for Increased Robustness Against Adversarial Attacks"
_ICLR.cc/2020/Conference — Accept (Poster)_

### Official Review · AnonReviewer3 · 2019-10-22
**Official Blind Review #3**

**Rating:** 6

**Review:**

The authors propose an ensemble of low-precision networks as a solution to providing a neural network with solid adversarial robustness whilst also providing good accuracy.

I found the paper easy to read with a high quality introduction and background, the results are very convincing and the idea is simple but intriguing. I think this will shift the community towards seriously considering low precision networks a partial solution to adversarial attacks (alongside adversarial training).

I could not work out from the paper whether the adversarial attacks on the low-precision networks were performed at full precision. I.e. someone could clone the low-precision networks, cast them to full precision, perform an adversarial attack like FGSM and then evaluate on the quantized network. It would be good to clarify this (or make it clearer in the text how you handle this).

**Experience Assessment:**

I do not know much about this area.

**Review Assessment: Checking Correctness Of Derivations And Theory:**

I did not assess the derivations or theory.

**Review Assessment: Checking Correctness Of Experiments:**

I assessed the sensibility of the experiments.

**Review Assessment: Thoroughness In Paper Reading:**

I made a quick assessment of this paper.

---

> ### Author Response · Authors · 2019-11-12
> **Response to Reviewer 3**
>
> We thank the reviewer for their positive comments. As correctly pointed out by the reviewer, this work was intended to showcase an alternative low-cost approach to increasing the robustness of deep learning models through the use of low-precision models, without sacrificing accuracy on the original unperturbed examples. Also, as discussed in our response to reviewer 1, other defense strategies like adversarial training, input gradient regularization, defensive distillation and full-precision ensembles suffer from limitations of increased training time, increased model size or increased inference time. However, the development of several hardware platforms and software libraries supporting low-precision operations has decreased the training and inference times for low-precision models allowing us to achieve increased robustness with minimal training, inference and model size overheads.
>
> We would like to clarify that the adversarial attacks on the low-precision models weren’t performed at full-precision. The attacked model was a low precision model utilizing quantized weights and activations. However, the gradients used in the attack generation were not quantized, allowing the adversary to launch a stronger attack. We have updated the paper to include this clarification.

---

### Official Review · AnonReviewer1 · 2019-10-24
**Official Blind Review #1**

**Rating:** 6

**Review:**

I think the paper reads well. It proposes to use ensembles of full precision and low-precision models in order to boost up robustness to adversarial attacks. It relies on the fact that low precision models are known to be more robust to adversarial attacks though performing poorly, while ensembling generally boosting up performance.

I think the premise of the paper is quite clear, and the results seem to be intuitive.  At a high level one worry that I have is if ICLR is the right conference for this work.

I would have expected maybe a more thorough empirical exploration. E.g. using resnets for ImageNet rather than AlexNet. Providing more baselines for the larger (and more reliable datasets) rather than MNIST which might be a bit misleading. I think the work does a decent job at looking at different number of components in the ensemble and analyzing the proposed method, but maybe not enough comparing and exploring other mechanism proposed as a defense for adversarial attacks.

However I think the message is clear, the results seem decent and I'm not aware of this being investigated in previous works.

**Experience Assessment:**

I have read many papers in this area.

**Review Assessment: Checking Correctness Of Derivations And Theory:**

I assessed the sensibility of the derivations and theory.

**Review Assessment: Checking Correctness Of Experiments:**

I assessed the sensibility of the experiments.

**Review Assessment: Thoroughness In Paper Reading:**

I made a quick assessment of this paper.

---

> ### Author Response · Authors · 2019-11-12
> **Response to Reviewer 1**
>
> We thank the reviewer for their comments. Please find the detailed responses to the individual concerns below.
>
>
> Suitability for ICLR:
>
> We believe that ICLR is the right venue for our paper for two main reasons. First, the high cost associated with ensemble models is often ignored by the machine learning community when considering its advantages in terms of increased performance and robustness. Their high memory and compute footprint can even be prohibitive on resource-constrained devices such as IoT edge devices and wearables. As an alternative, we propose mixed-precision ensembles and illustrate their advantages in terms of both higher robustness and low compute and memory overhead. Second, over the past few years, ICLR has published many papers on low precision networks [1][2]. This work builds on the existing works and demonstrates an additional advantage of these low precision networks.
>
> [1] Aojun Zhou et al. “Incremental Network Quantization: Towards Lossless CNNs with Low-Precision Weights”. ICLR 2017.
> [2] Angus Galloway et al. “Attacking Binarized Neural Networks.” ICLR 2018.
>
>
> Additional baselines for benchmarks other than MNIST:
>
> We did not present additional baselines for the CIFAR-10 and AlexNet benchmarks as they did not yield networks with higher adversarial accuracies and <5% drop in unperturbed accuracies compared to the full-precision baselines. To illustrate the point further, we present the results for CIFAR-10 with defensive distillation and input gradient regularization below. The distillation process was implemented with a softmax temperature of T = 100, the gradient regularization was realized with a regularization penalty of lambda = 200.
>
> +++++++++++++++++++++++++++++++++++++++++++++++++++++++++++++++++++++++++++++++++++
> Defense strategy  |  Unperturbed Accuracy |  CW   | FGSM   |   BIM  |  PGD  | Average Adversarial
> +++++++++++++++++++++++++++++++++++++++++++++++++++++++++++++++++++++++++++++++++++
> Defensive distill.   |          	63.84                     |  31.4  |   14.4   |   5.83  |   4.08  |          13.93
> Inp. Grad. Reg.      |          	74.91                     | 12.58 |  10.06  | 12.72  | 10.43  |          11.45
>
>
> Comparison with other mechanisms proposed as a defense for adversarial attacks:
>
> Among the plethora of works on increasing robustness, we have chosen to compare our work with some of the most cited and popular defense strategies, namely, FGSM based adversarial training, defensive distillation and input gradient regularization, due to page restrictions and implementation efforts. However, as requested by Reviewer 4, we have also updated the paper to include the comparison with PGD-based adversarial training. We would like to highlight that our approach stands out from previous work in terms of drastically lower overheads. Adversarial training, input gradient regularization and defensive distillation all increase the overall training time significantly, while ensembling with full-precision models increase the overall model size and inference time several-fold. In contrast, with the development of hardware that natively supports low-precision operations and the development of libraries that can take advantage of these low precision computation engines (Ex: CUDA 10 on Turing GPUs https://devblogs.nvidia.com/cuda-10-features-revealed/), the training and inference times for low-precision models are decreasing remarkably (https://devblogs.nvidia.com/int4-for-ai-inference/). We exploit this advantage of low-precision models to achieve increased robustness with minimal increases in overall training and inference times (training and executing two low precision models in addition to one full-precision model).

---

### Official Review · AnonReviewer4 · 2019-10-30
**Official Blind Review #4**

**Rating:** 1

**Review:**

This paper suggests using ensemble of both full-precision and low-bits precision models to defense adversarial examples.

From methodological point of view, this idea is quite straightforward and not novel, since there are already several works that applied ensemble methods to improve the robustness of NNs, including the Strauss et.al 2017 and (the following references are not included in the manuscript)
"Adversarial Example Defenses: Ensembles of Weak Defenses are not Strong
Warren He, James Wei, Xinyun Chen, Nicholas Carlini, Dawn Song"
"Ensemble Adversarial Training: Attacks and Defenses
Florian Tramèr, Alexey Kurakin, Nicolas Papernot, Ian Goodfellow, Dan Boneh, Patrick McDaniel" .
"Improving Adversarial Robustness via Promoting Ensemble Diversity
Tianyu Pang, Kun Xu, Chao Du,  Ning Chen,  Jun Zhu " ICML 2019

Though these methods only considered combining full-precision models, the idea is the same in essence and let the low-bits networks involve into the ensemble is quite natural and straightforward. So I don't think the methodology contribution of this paper is enough for publication.

When checking the empirical results, the compared baselines miss a very common-used and strong baseline PGD adversarial training. And also the performance of this ensemble is not significant.

Considering the weakness of the paper both in methodology development and empirical justification, this work does not merit publication from my point of view.

**Experience Assessment:**

I have published in this field for several years.

**Review Assessment: Checking Correctness Of Derivations And Theory:**

I assessed the sensibility of the derivations and theory.

**Review Assessment: Checking Correctness Of Experiments:**

I assessed the sensibility of the experiments.

**Review Assessment: Thoroughness In Paper Reading:**

I read the paper at least twice and used my best judgement in assessing the paper.

---

> ### Author Response · Authors · 2019-11-12
> **Response to Reviewer 4**
>
> We thank the reviewer for their comments. Please find the detailed responses to the individual concerns below.
>
>
> Related efforts on ensembles and unique contribution of this work:
>
> We thank the reviewer for pointing out the additional related work; we have updated the related work section to include these works. However, we feel that our work makes a significant contribution above the previous work when it comes to computationally-efficient defenses. Previous approaches either increase the training time greatly (adversarial training, input gradient regularization and defensive distillation), or increase the inference memory and compute footprint several-fold (full-precision ensembles). As a result, these approaches are inapplicable to resource-constrained systems (like IoT edge devices and wearables). This is the problem addressed in our work.
>
> Recent years have seen a tremendous growth in efforts towards DNNs optimized for computational efficiency [1] [2]. Following the same motivation, this work demonstrates a computationally-efficient approach of utilizing mixed-precision ensembles to increase robustness while maintaining unperturbed accuracy. Advances in hardware that natively supports low-precision operations and software libraries that can take advantage of these low precision computation engines (Ex: CUDA 10 on Turing GPUs https://devblogs.nvidia.com/cuda-10-features-revealed/) infact allow low-precision models to execute much faster than their full-precision counterparts (https://devblogs.nvidia.com/int4-for-ai-inference/), thereby restricting the inference time overheads of EMPIR to <25%. Further, unlike other popular non-ensemble defense techniques like adversarial training, input gradient regularization and defensive distillation, our approach doesn’t increase training time. The overall idea is simple, but effective. We believe that its successful implementation, as demonstrated in this paper, is an important step towards realizing computationally efficient and robust DNNs.
>
> [1]  A.G Howard et al. “MobileNets: Efficient Convolutional Neural Networks for Mobile Vision Applications”. ArXiv, abs/1704.04861 (2017).
> [2], Forrest N. Iandola et al. “SqueezeNet: AlexNet-level accuracy with 50x fewer parameters and <1MB model size.” ArXiv abs/1602.07360 (2017).
>
>
> Comparison with PGD adversarial training:
>
> Since there is a plethora of efforts on increasing robustness, we restricted the comparisons to a few popular representative works due to space and time restrictions. FGSM adversarial training results were presented instead of PGD adversarial training because it converges faster. However, as requested by the reviewer, we have updated the paper to include the following results on PGD adversarial training [3], and we will include additional results in the final paper. The adversarial training was performed on adversarial examples generated with a maximum possible perturbation of epsilon = 0.3.
>
> ++++++++++++++++++++++++++++++++++++++++++++++++++++++++++++++++++++++++++++++++++++++++++++++++
> Network       |         Approach       | Unperturbed Accuracy |    CW   |   FGSM   |   BIM   |  PGD  | Average Adversarial
> ++++++++++++++++++++++++++++++++++++++++++++++++++++++++++++++++++++++++++++++++++++++++++++++++
> CIFARconv    | PGD Adv. Train      |                  73.55               |  12.62   |  12.45   |  10.97  |  8.52  |            11.14
> CIFARconv    | EMPIR                     |                   72.56              |   48.51  |   20.61  |  24.59  | 13.34 |             26.76
>
> As evident from the above values, for the CIFARconv benchmark, EMPIR achieves a higher average adversarial accuracy than PGD adversarial training. EMPIR is able to achieve this improvement with zero training overhead, whereas PGD adversarial training increases the training time significantly because of the need to construct adversarial examples during training (One PGD adversarial training epoch is 22x slower than a clean training epoch on an RTX 2080 Ti GPU).
>
> [3] A. Madry et al. “Towards Deep Learning Models Resistant to Adversarial Attacks.” ArXiv abs/1706.06083 (2017).

---

### Decision · Program_Chairs · 2019-12-19

**Decision:**

Accept (Poster)

**Comment:**

This paper proposed to apply emsembles of high precision deep networks and low precision ones to improve the robustness against adversarial attacks while not increase the cost in time and memory heavily.  Experiments on different tasks under various types of adversarial attacks show the proposed method improves the robustness of the models without sacrificing the accuracy on normal input.  The idea is simple and effective.  Some reviewers have had concerns on the novelty of the idea and the comparisons with related work but I think the authors give convincing answers to these questions.